# The Long-Term Efficacy, Prognostic Factors, Safety, and Hospitalization Costs Following Denervation and Myotomy of the Affected Muscles and Deep Brain Stimulation in 94 Patients with Spasmodic Torticollis

**DOI:** 10.3390/brainsci12070881

**Published:** 2022-07-04

**Authors:** Zhiqiang Cui, Tong Chen, Jian Wang, Chao Jiang, Qingyao Gao, Zhiqi Mao, Longsheng Pan, Zhipei Ling, Jianning Zhang, Xuemei Li

**Affiliations:** 1Department of Neurosurgery, The First Medical Clinical Center, PLA General Hospital, Beijing 100853, China; zhiqiangcui2008@hotmail.com (Z.C.); wj251301@126.com (J.W.); markmaoqi@126.com (Z.M.); panls301@163.com (L.P.); lzp301hos@126.com (Z.L.); zhangjn301hos@126.com (J.Z.); 2Department of Neurology, The Second Medical Center & National Clinical Research Center for Geriatric Diseases, Chinese PLA General Hospital, Beijing 100853, China; ct301hos@126.com; 3School of Basic Medical Sciences, North China University of Science and Technology, No. 21 Bohai Road, Caofeidian Eco-City, Tangshan 063210, China; jiangchao103@163.com (C.J.); qingyaogao321654@163.com (Q.G.); 4Cadre Medical Department, The First Medical Clinical Center, PLA General Hospital, 28 Fuxing Road, Haidian District, Beijing 100853, China

**Keywords:** spasmodic torticollis (ST), deep brain stimulation (DBS), denervation and myotomy (DAM), hospitalization cost, prognostic factors

## Abstract

The surgical methods for treating spasmodic torticollis include the denervation and myotomy (DAM) of the affected muscles and deep brain stimulation (DBS). This study reports on the long-term efficacy, prognostic factors, safety, and hospitalization costs following these two procedures. We collected data from 94 patients with spasmodic torticollis, of whom 41 and 53 were treated with DAM and DBS, respectively, from June 2008 to December 2020 at the Chinese People’s Liberation Army General Hospital. We used the Tsui scale and the global outcome score of the Toronto Western Spasmodic Torticollis Rating Scale (TWSTRS) to evaluate the preoperative and postoperative clinical conditions in all patients. We also determined the costs of hospitalization, prognostic factors, and serious adverse events following the two surgical procedures. The mean follow-up time was 68.83 months (range = 13–116). Both resection surgery and DBS showed good results in terms of Tsui (Z = −5.103, *p* = 0.000; Z = −6.210, *p* = 0.000) and TWSTRS scores (t = 8.762, *p* = 0.000; Z = −6.308, *p* = 0.000). Compared with the DAM group, the preoperative (47.71, range 24–67.25) and postoperative (18.57, range 0–53) TWSTRS scores in the DBS group were significantly higher (Z = −3.161, *p* = 0.002). We found no correlation between prognostic factors and patient age, gender, or disease duration for either surgical procedure. However, prognostic factors were related to the length of the postoperative follow-up period in the DBS surgery group (Z = −2.068, *p* = 0.039; Z = −3.287, *p* = 0.001). The mean hospitalization cost in the DBS group was 6.85 times that found in the resection group (Z = −8.284, *p* = 0.000). The total complication rate was 4.26%. We found both resection surgery and DBS showed good results in the patients with spasmodic torticollis. Compared with DAM, DBS had a greater improvement in TWSTRS score; however, it was more expensive. Prognostic factors were related to the length of the postoperative follow-up period in patients who underwent DBS surgery.

## 1. Introduction

Spasmodic torticollis (ST) is a chronic neurological disorder in which the head turns or tilts in jerky movements or sustains a prolonged and awkward position due to the involuntary contractions of neck muscles. ST is associated with neck pain, significant restrictions in daily activities, and reduced social participation and thus has a profound effect on quality of life [1]. Oral treatment is often ineffective or limited by accompanying side effects, and botulinum neurotoxin (BoNT) injections are the treatment of choice [2]. However, around 25% of patients do not benefit from BoNT (primary nonresponders) or lose the treatment benefit after an initial response (secondary nonresponders) [3]. For patients with treatment-refractory ST, the primary surgical options include the denervation and myotomy (DAM) of the associated muscles and deep brain stimulation (DBS). Although both treatments are effective, they carry high hospitalization costs in China. Thus, economic constraints must be considered when choosing a surgical treatment method. This study reports on the efficacy, prognostic factors, safety, and hospitalization costs of the two procedures.

## 2. Methods

### 2.1. Patients

We collected data from 94 patients with ST who underwent DAM or DBS from June 2008 to December 2020. Although DAM was previously the primary surgical therapy used to treat ST, it was surpassed by DBS in 2012. ST is clinically classified according to neck posture, and the types include rotational torticollis, laterocollis, retrocollis, anterocollis, and various combinations of these. There were no patients with pure anterocollis in the present study. Four cases were diagnosed as tardive cervical dystonia. All patients underwent pharmacotherapy, and some patients also underwent psychotherapy and physiotherapy. Cervical botulinum toxin injections were given in 68 patients. In total, we analyzed data from 52 men and 42 women. The mean age was 46.82 years (±12.47, range 16–72), the mean duration of symptoms was 44.73 months (±47.84, range 3–252), and the mean follow-up period was 68.83 months long (±37.53, range 13–146). DAM was performed in 41 patients, and 53 patients underwent DBS (Table 1).

### 2.2. DAM Techniques

Forty-one patients underwent DAM techniques. The triad procedure [4,5] was used in 19 patients who had mainly rotational symptoms, and included (1) denervation of the ipsilateral posterior cervical group through extradural section of the roots of C1 and C2, combined with section of the posterior primary divisions (ramisectomy) of C3, C4, and C5 [6]; (2) resection of the splenius capitis/cervicis or levator scapulae on the side towards which the chin turned [7]; and (3) denervation of the contralateral sternocleidomastoid (SCM) with transection of the muscle [8]. In 14 patients with laterocollis and partial rotational symptoms, extradural section of the roots of C1 and C2 was performed, combined with section of the posterior primary divisions (ramisectomy) of C3, C4, and C5 and denervation of the contralateral SCM with transection of the muscle. In four patients with rotational torticollis and laterocollis, ipsilateral microvascular decompression of the accessory nerve [9] was performed with subdural selective C3–5 sensory nerve root rhizotomy. Two patients with only rotational symptoms underwent denervation of the contralateral SCM with transection of the muscle. Two patients with retrocollis underwent bilateral myotomies of the splenius capitis/cervicis and semispinalis capitis/cervicis (Table 2). No patients had anterocollis.

### 2.3. DBS Surgical Procedure

Fifty-three patients underwent DBS. One or two days prior to surgery, MRI was performed on all patients using a 1.5 or 3.0 T scanner (MAGNETOM Espree; Siemens Healthineers, Erlangen, Germany). On the day of surgery, a model F stereotactic head frame (Leksell; Elekta, Stockholm, Sweden) was positioned on the patient and a computed tomography (CT) scan was performed. The CT images were fused with the pre-surgical MRI, and preoperative planning was conducted to determine the optimal entry points for a safe electrode trajectory and avoidance of blood vessels (especially blood vessels within the sulcus) and ventricles. A Robotic Stereotactic Assistance (ROSA, MedTech Surgical, Inc., Montpellier, France) device was used to guide the surgical procedure in a minority of patients. Intraoperative microelectrode recording (MER) was routinely performed in most patients. Single-channel MER or multi-channel MER was generally performed using intraoperative microelectrodes (Medtronic Inc., Minneapolis, MN, USA) and the Leadpoint Neural Activity Monitoring System (Medtronic Inc.), although an Alpha Omega recording system (Alpha Omega, Nazareth, Israel) was used in a minority of patients. The electrodes were advanced to 8 mm above the target using a clinical microdrive (microTargeting Drive; FHC Inc., Bowdoin, ME, USA). After MER, the DBS electrodes (Medtronic Inc. and Beijing PINS Medical Co.) were guided into place. General anesthesia was used during the entire surgical procedure. After implantation of the bilateral intracerebral electrodes, an intraoperative MRI scan (Siemens Espree, 1.5 T) was immediately performed. The main purpose of this iMRI was to assess the accuracy of the electrode positioning, which was confirmed using a fused image of the intraoperative 3D T1-weighted sequence and the preoperative plan. The intraoperative MRI was also used to examine the possibility of intraoperative bleeding, infarction, and intracranial gas accumulation (pneumocranium).

In most patients, the implantation target was the globus pallidus internus (GPi). The anatomical coordinates of the target posteroventral GPI was 3 mm anterior to the midcommissural point, 18–21 mm lateral to the midplane of the third ventricle, and 4–6 mm below the intercommissural line. In two patients who had ST combined with Parkinson’s disease (limb tremor), electrodes were implanted in bilateral subthalamic nucleus (STN) targets; in one patient with ST combined with intense head tremors, electrodes were implanted in bilateral VIM targets; and in one patient with ST combined with Parkinson’s disease (limb tremors and stiffness), four electrodes with two rechargeable pulse generators were implanted in bilateral STN and GPi targets. For stimulation parameters of GPI, amplitude, pulse width, and frequency were 2.2–4.0 V, 60–120 μsec, and 130–180 Hz, respectively.

In one patient (Patient 2), the Activa PC (37,603) was replaced by an Activa RC (37,612, rechargeable battery) at postoperative 5 years and one month because of battery energy depletion. Rechargeable pulse generators were implanted in 21 patients. Eight patients chose the Medtronic Inc. device and 24 patients chose the Beijing PINS Medical Co. device (Beijing, China) (Table 2).

Patient clinical condition was evaluated using the Tsui scale [10] and the global outcome score of the Toronto Western Spasmodic Torticollis Rating Scale (TWSTRS) [11]. A neurologist who was experienced in movement disorders performed these assessments shortly before surgery and at the last follow-up. We also tabulated the hospitalization costs and the serious adverse events associated with the two surgical procedures.

### 2.4. Statistical Analysis

We used SPSS^®^ Statistics Version 24.0 (IBM Corp., Armonk, NY, USA) for statistical analyses. Measurement data are expressed as (x ± s). Two independent sample t-tests were used for comparisons between groups, and paired t-tests were used for comparisons within groups. The count data are expressed as % and the chi-squared test was used for categorical variables. The Wilcoxon signed rank sum test was also used to analyze the data. *p* < 0.05 was considered statistically significant.

## 3. Results

Although the patient treatments were not randomized, there were no significant differences in gender, age, or the duration of symptoms before surgery between the DBS group and the DAM group (*p* = 0.404; *p* = 0.073; *p* = 0.740) (Table 1).

Compared with the DBS group, the follow-up time was longer in the DAM group (*p* = 0.000). This is because the DAM technique was the primary surgical therapy for ST before 2012, and DBS has since replaced DAM as the primary surgical therapy. The medical expenses were lower in the DAM group (*p* = 0.000). The mean hospitalization costs were USD 5934.03 (range = USD 3023.38–USD 11,940.74) in the DAM group and USD 40,636.54 (range = USD 25,192.83–USD 69,990.43) in the DBS group, such that the cost of the treatment in the DBS group was 6.85 times that of the DAM group (Table 1) (as of 31 December 2020, CNY 1 equals USD 0.1528).

In terms of the TWSTRS score, we found no significant differences between the two groups before surgery (*p* = 0.650). The mean preoperative total TWSTRS score was 48.81 (range = 30–68) in the DAM group and 47.71 (range 24–67.25) in the DBS group. The mean postoperative total TWSTRS score was 28.21 (range = 0–64.25) in the DAM group and 18.57 (range = 0–53) in the DBS group. The postoperative scores improved significantly in both groups (*p* = 0.000). The mean improvement in the postoperative TWSTRS score was larger in the DBS group (61.08%) than in the DAM group (42.2%) (*p* = 0.002) (Table 3, Figure 1).

The Tsui score was higher in the DBS group before surgery (*p* = 0.011), and the postoperative Tsui scores improved significantly in both groups (*p* = 0.000). The mean preoperative Tsui score was 8.61 (range = 2–19) in the DAM group and 10.02 (range = 2–17) in the DBS group. The mean postoperative Tsui score was 4.68 (range = 0–14) in the DAM group and 3.62 (range = 0–13) in the DBS group. The mean improvement in the postoperative Tsui score was not significantly different: 45.64% in the DAM group and 63.87% in the DBS group (*p* = 0.072) (Table 3, Figure 2).

In the DAM group, the preoperative TWSTRS score was higher in patients with a short disease duration (less than 24 months) compared with those with a disease duration longer than 24 months (z = −3.800, *p* = 0.000). This indicates that in the DAM group, a shorter disease history was associated with more severe symptoms. In terms of prognosis, we found no significant correlations between gender, age at surgery, disease duration, the length of the follow-up period, or postoperative curative effect (Table 4).

In the DBS group, correlation analyses showed that the preoperative TWSTRS scores in female patients were higher than those in men (z = −2.119, *p* = 0.034), indicating that women had more serious symptoms. Furthermore, the preoperative Tsui scores in patients over 50 years old were higher than in those younger than 50 years. This indicated that older patients in the DBS group had more serious symptoms, such as the degree and prevalence of head tilt and the occurrence of tremor (z = −2.930, *p* = 0.003). In terms of prognosis, we found no significant correlations between patient gender and age at surgery, disease duration, or postoperative curative effect. However, the prognosis was related to postoperative follow-up time such that patients with a longer period of postoperative stimulation had significantly greater reductions in symptoms. Specifically, the TWSTRS score (z = −2.068, *p* = 0.039) and Tsui score were significantly decreased (z = −3.287, *p* = 0.001) in patients for whom the follow-up period was longer than 48 months (Table 5).

After receiving DAM treatment at other hospitals, three patients were not satisfied with the postoperative outcome for more than 2 years and therefore elected to undergo DBS in our hospital. There were no patients who received DAM after DBS. Only seven patients had been treated with botulinum toxin in a short time after operation, five in the DAM group and two in the DBS group.

No deaths were registered among the entire patient group, and the total complication rate was 4.26% (4 out of 94 patients). Intraoperative bleeding occurred in one patient. Specifically, a fresh blood outflow conduit formed during the DBS procedure. The operation was immediately suspended, and strict blood pressure control was applied along with hemostatic agents. An intraoperative MRI revealed a large amount of hemorrhaging (about 20 mL). To address this, a drainage tube was implanted via stereotactic guidance in the hematoma through another bone hole, and the patient was immediately subjected to intraoperative hematoma aspiration. With the consistent maintenance of the drainage, the patient had only mild hemiplegia, representing a good outcome (Figure 3). In another patient, the intraoperative MRI revealed ischemia in the left frontal lobe. Symptoms in the acute phase included headache, nausea, partial aphasia, cognitive changes, and hallucinations, although there were no long-term complications (Figure 4). One patient suffered from an infection at the neck incision after the triad procedure; this was resolved via debridement. Finally, one patient suffered from moderate quadriplegia after the subdural selective C3–5 sensory nerve root rhizotomy. A CT scan showed a postoperative epidural hematoma in the surgical area. This hematoma was removed, and limb muscle strength was basically restored afterwards such that the patient had no permanent hemiplegia.

## 4. Discussion

Both DAM and DBS had good results in our patients with ST (*p* < 0.001). The DAM procedure is performed in only a few neurosurgical departments worldwide, and other authors have reported similar results [12,13]. For instance, Bertrand et al. [6,7,14] reported excellent or good results in 88% of patients who underwent DAM treatment. Further, Meyer [15] reported that in patients with ST who were resistant to injections of botulinum toxin, ramisectomy led to improved TWSTRS scores. This was the case in 12 of their 14 patients (85.7%), and the mean follow-up duration was 36.3 months, with about one-third of the patients experiencing modest long-term functional improvements. Braun et al. [16] reported that in 155 patients with ST who underwent selective peripheral denervation, after a mean follow-up period of 32.8 months, 73% of the patients were satisfied with the outcome of the operation.

In our patient group, TWSTRS and Tsui scores improved by an average of 42.20% (range = 0–100) and 45.64% (range = 0–100), respectively, following the DAM surgery. Thus, the outcome in our patient group was less favorable than that reported in the literature. Wang et al. [4] examined data from a large sample of 648 patients with ST who underwent selective peripheral denervation. They found a statistically significant improvement between the preoperative and postoperative TWSTRS evaluation of 73.5 ± 11.9% during a mean follow-up period of 33.4 months. Differences in reported treatment outcomes may be related to differences in the type of ST, surgical methods used, degree of experience of the surgeon, operation duration, and follow-up time. Unfortunately, our study included a relatively small number of cases (41 patients) and diverse surgical procedures such as the triad procedure, selective peripheral denervation, microvascular decompression of the accessory nerve, transection of the sternocleidomastoid, and myotomies of the splenius capitis/cervicis and semispinalis capitis/cervicis.

GPi-DBS has been found to be an effective and relatively safe treatment for patients with severe generalized or segmental dystonia [17,18]. Kiss et al. [19] conducted a prospective, single-blind, multicenter study assessing the efficacy and safety of bilateral GPi-DBS in 10 patients with severe, chronic, medication-resistant cervical dystonia and reported a beneficial effect in cervical dystonia. Another prospective, randomized, multicenter study showed that 3 months of GPi-DBS in 62 patients with cervical dystonia was more effective than sham stimulation in terms of symptom reduction [20]. Other authors have reported that DBS can improve the total TWSTRS score by about 54–76% [21,22,23,24,25,26,27]. In our patient group, the mean improvement in the postoperative TWSTRS score was larger in the DBS versus DAM group (Z = −3.161, *p* = 0.002). Specifically, the TWSTRS total score was reduced by 61.08% (Z = −6.308, *p* = 0.000) in the DBS group. However, there were no significant differences (Z = −1.796, *p* = 0.072) in the mean postoperative Tsui improvement score. This may be because the Tsui score only assesses the degree of head and neck tilt, shoulder lift, duration of head tilt, and head tremor, while the TWSTRS assesses these factors as well as quality of life and degree of pain relief. Improvements in quality of life and pain relief are most closely aligned with the needs of ST patients.

Few studies have reported on the prognostic factors related to the surgical efficacy of ST treatments, especially those related to resection surgery. In the DAM group, we found that patients with disease durations of less than 24 months had more severe symptoms than those with a longer disease duration. However, in terms of prognosis, there were no significant correlations between patient gender and age at surgery, disease duration, the length of the follow-up period, and the postoperative curative effect.

Regarding the prognostic factors related to DBS surgery outcome, Hua’s meta-analysis showed that the effect of DBS on ST was related to the age at surgery, such that patients who had a younger age of onset were more likely to have a good postoperative outcome. Further, there was no correlation between the surgery outcome and gender, medical history, or follow-up time [27]. However, our results showed no significant correlation between the age at surgery, gender, disease duration, and postoperative curative effect, although the prognosis was related to the length of the postoperative follow-up period. Specifically, symptoms were significantly lessened in patients who continued long-term postoperative stimulation for more than 48 months. The differences between our results and those of Hua may be related to our small sample size, single-center design, and retrospective analysis. The long-term postoperative stimulation for more than 48 months obtained a better postoperative outcome; our results indicate that the long-term stimulation of GPI-DBS can regulate the network disorder of cervical dystonia and reshape brain network function.

Which of the two surgical methods (DBS and DAM) is preferable for patients with ST? Francesco et al. compared GPi-DBS with resection surgery in patients with ST and seemed to favor the former. In their study, the DBS procedure had a better outcome in terms of long-term efficacy and was rated as more satisfactory by patients in terms of pain relief [24]. Furthermore, Huh et al. [28] examined 24 patients with cervical dystonia who underwent DBS and DAM and found that the two procedures had similar efficacies, although there was a trend toward greater pain reduction in the DBS group (*p* = 0.094). Fiorella et al. compared selective peripheral denervation (20 patients) with pallidal stimulation (15 patients) for the treatment of cervical dystonia and concluded that GPi-DBS was superior because it had a larger benefit, even if it carried a greater risk of severe complications [29]. GPi-DBS is a valid option in the case of failed resection surgery [29]. Taken together, the previous literature indicates that the improvement rate of resection surgery is about 73–88% [4,6,7,12,13,14,16], while that of DBS is about 54–76% [21,22,23,24,25,26,27]. However, while the former had better overall efficacy, DBS led to better results in terms of pain relief and long-term efficacy. The number of DBS cases is still small, and double-blind, randomized, controlled studies are needed. Thus, the optimal surgical procedure is still inconclusive.

Why can DBS better improve the clinical symptoms of patients with ST? We infer the following factors: (1) DBS regulates the brain network, the cortex-striatum-thalamus-cortex loop, so all the peripheral nerves in the neck are regulated, while ST only involves multiple muscles of the neck; DMA can only remove the limited muscles and nerves of the neck, and the deviation of the neck involves more muscles, not a few muscles; (2) DBS is a long-term neuromodulation, the parameters can be adjusted, the clinical effect will be better, and the brain function can be reconstructed. However, for DMA surgery, cervical nerve reinnervation in 1–2 years may lead to the recurrence of symptoms; (3) the etiology of ST is brain dysfunction. DBS directly acts on the cortical basal ganglia circuit, which is the etiological treatment, while DMA is the symptomatic treatment, and the removed muscles and nerves are normal.

In terms of sex ratio, there is an anomaly in our results. There were 52 males and 42 females, more male patients than female patients, which is different from the overall prevalence of isolated cervical dystonia, which typically has much higher proportions of females. That is because we only counted patients who have undergone surgery, not the incidence rate of isolated cervical dystonia. Whether patients can receive surgical treatment is related to economic conditions and traditional concepts in China. This is why the article analyzes the hospitalization expenses of patients. In developing countries, economic factors have an important impact on whether patients can take reasonable treatment.

To the best of our knowledge, no previous reports have examined the cost of selective peripheral denervation in ST patients. In this study, we found that the mean medical costs were USD 5934.03 in the DAM group. The patient with the highest costs (USD 11,787.66) underwent a subdural selective C3–5 sensory nerve root rhizotomy and then developed a postoperative epidural hematoma, which required a second operation, the rehabilitation of limb function, and a prolonged hospital stay.

In terms of the cost of DBS, Lad et al. [30] utilized the Nationwide Inpatient Sample (NIS) database and reported that among 34,792 patients who underwent DBS surgery from 1993 to 2006, the average cost of DBS surgery gradually increased from USD 38,840 in 1993 to USD 69,329 in 2006. Eskandar et al. [31] examined the records of a representative sample of US hospitals and reported that the median total hospital charge for 589 patients who received a neurostimulator device was USD 35,700 between 1996 and 2000. Sharma et al. [32] conducted a retrospective cohort study using the United States NIS. They found that from 2006 to 2010, 2228 patients with Parkinson’s disease underwent intracranial neurostimulator implantation. In this group, the mean total cost of hospitalization increased from USD 56,534.30 in 2006 to USD 59,875.27 in 2010, with an overall mean expenditure of USD 61,362.93 ± USD 36,016.21.

The present study is the first to report on the cost of DBS in China. We found that the mean cost of DBS was USD 40,636.54 (range = USD 24,869.85–USD 69,093.13). Forty-two (79.25%) patients received rechargeable pulse generators, which were more expensive. A patient with four electrodes and two rechargeable pulse generators had a high cost (USD 69,093.13), as did a patient who had intraoperative bleeding during the DBS procedure (USD 62,106.75) and had to pay extra for drainage tube implantation, the treatment of hypostatic pneumonia, the rehabilitation of limb function, and a prolonged hospital stay. Compared with previous literature, the mean total cost of hospitalization was lower in our patient group. However, compared with the DAM group, the costs associated with DBS were almost 6.85 times higher (*p* < 0.001). Although no previous studies have compared hospitalization costs between DBS and DAM in ST patients, we expect that the hospitalization costs of DBS are routinely higher because of the cost of the procedure and implanted device. These implanted devices are not covered by medical insurance in most parts of China, and so the surgical treatment of ST via DBS undoubtedly carries an increased financial burden for people in developing countries. Thus, when choosing a surgical method to treat ST, patients and physicians must consider efficacy, safety, and also the patient’s financial situation.

In recent years, most authors seem to favor DBS for treating ST. However, DBS-related complications, such as surgery-related bleeding or infarction, hardware-related infection, lead fracture, lead migration, stimulation-related symptoms, are not uncommon. Although there were no fatalities in our patient group, one patient experienced a hemorrhage, resulting in permanent moderate hemiplegia, and another patient developed ischemia in the left frontal lobe, with no long-term complications. There were no hardware-related complications or stimulation-related symptoms, potentially because of the small number of cases. There were no serious complications associated with DAM, likely because it does not interfere with the central nervous system; one patient who underwent the subdural procedure had temporary quadriplegia because of epidural hematoma, but there was no permanent hemiplegia.

## 5. Conclusions

Both DAM and DBS showed good results in our group of patients with ST. Compared with DAM, DBS had a more improvement in TWSTRS score, although the hospitalization cost was substantially higher. The choice of the surgical method for ST in developing countries should be individualized and should consider both efficacy and affordability. The outcome of DBS was not significantly correlated with age at surgery, gender, disease duration, or postoperative curative effect but was related to the postoperative follow-up duration.

## Figures and Tables

**Figure 1 brainsci-12-00881-f001:**
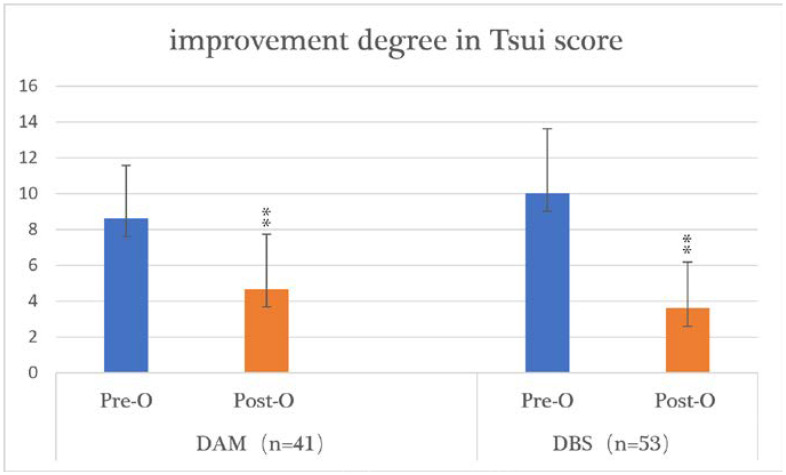
The improvement in Tsui scores in the DAM group and DBS group. TWSTRS: the Toronto Western Spasmodic Torticollis Rating Scale; DAM: denervation and myotomy; DBS: deep brain stimulation; Pre-O: preoperative; Post-O: postoperative; **: *p* < 0.01.

**Figure 2 brainsci-12-00881-f002:**
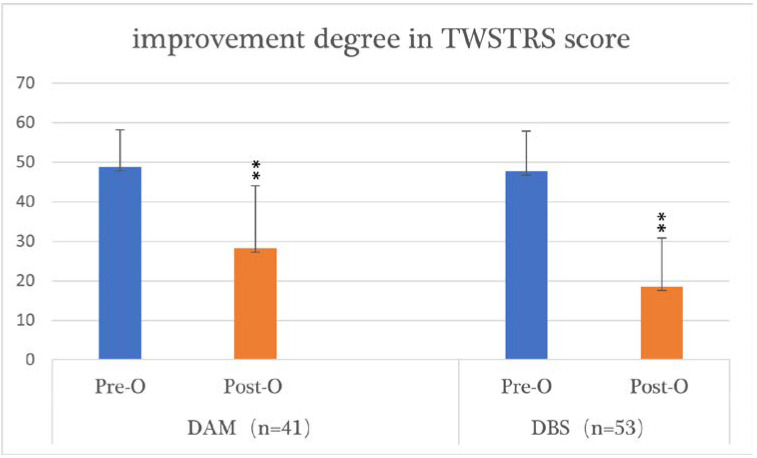
The improvement in TWSTRS scores in the DAM group and DBS group. TWSTRS: the Toronto Western Spasmodic Torticollis Rating Scale; DAM: denervation and myotomy; DBS: deep brain stimulation; Pre-O: preoperative; Post-O: postoperative; **: *p* < 0.01.

**Figure 3 brainsci-12-00881-f003:**
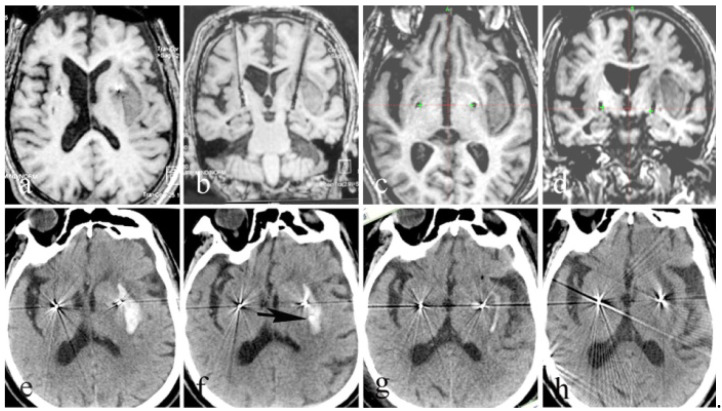
Globus pallidus internus DBS. (**a**–**d**): Axial and coronal T1-weighted intraoperative MR images (**a**,**b**). Intraoperative MR images were fused with preoperative MR images. The green “+” (**c**,**d**) indicates that the electrodes did not shift, despite a large hematoma. (**e**–**h**): A puncture trajectory was designed according to the intraoperative MR data, with the hematoma as the target. A drainage tube (black arrow) was placed in the hematoma (**f**). Several days later, CT showed that the hematoma had completely drained and that the electrode position was acceptable (**g**,**h**).

**Figure 4 brainsci-12-00881-f004:**
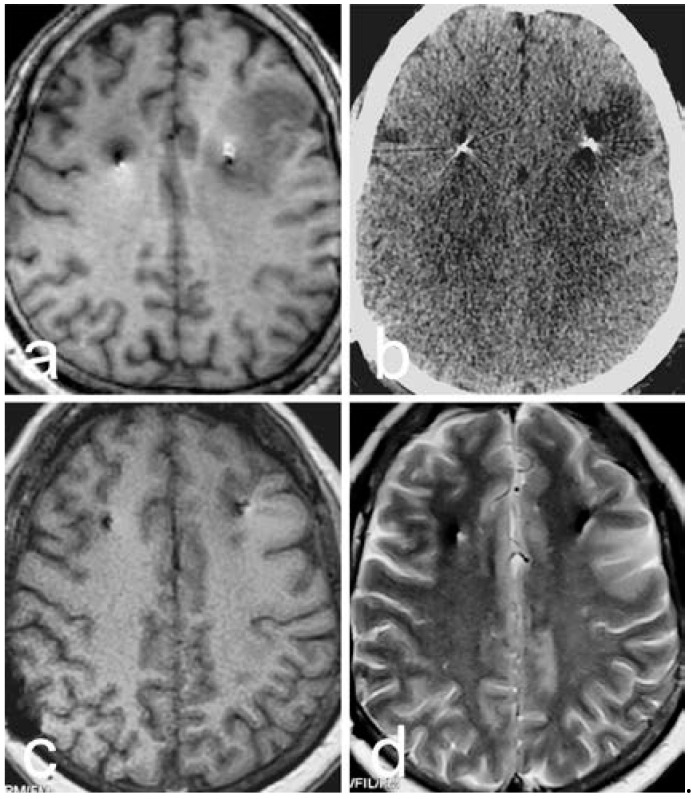
DBS of the globus pallidus internus. (**a**): The intraoperative 3D T1-weighted image shows low signal around the left electrode. (**b**): A CT scan conducted on 1 day postoperation shows the low density of the ischemic infarct. (**c**,**d**): T1- and T2-weighted MRI images at 9 months postoperation show a slight abnormality around the leads.

**Table 1 brainsci-12-00881-t001:** The characteristics of 94 patients treated for ST via surgical procedures.

	DAM (*n* = 41)	DBS (*n* = 53)	t/χ2	*p*-Value
Sex			χ2 = 0.941	*p* = 0.404
M	25 (61.0%)	27 (50.9%)		
F	16 (39.0%)	26 (49.1%)		
Age (Y)	49.44 ± 11.54	44.79 ± 12.88	t = 1.814	*p* = 0.073
DOS (M)	41.66 ± 46.64	47.11 ± 49.06	z = −0.332	*p* = 0.740
FU (M)	105.46 ± 18.02	40.49 ± 19.82	z = −8.072	*p* < 0.001 *
ME (USD)	5934.03 ± 2086.95	40,636.54 ± 1046.88	z = −8.284	*p* < 0.001 *

DAM: denervation and myotomy; DBS: deep brain stimulation; DOS: duration of symptoms; Y: year; M: month; FU: follow-up; ME: medical expenses; USD: US dollar; *: *p* < 0.05.

**Table 2 brainsci-12-00881-t002:** Surgical procedures in 94 patients with ST.

Surgical Procedure		Case
DAM	Triad procedure [4,5]	19
	Extradural section of the roots of C1 and C2, combined with section of the posterior primary divisions (ramisectomy) of C3, C4 and C5 + denervation of contralateral SCM with a transection of the muscle	14
	Ipsilateral microvascular decompression of the accessory nerve [9] + subdural selective C3-5 sensory nerve root rhizotomy	4
	Denervation of contralateral SCM with a transection of the muscle	2
	Bilateral myotomies of the splenius capitis/cervicis and semispinalis capitis/cervicis	2
DBS	GPI-DBS	49
	STN-DBS	3
	VIM-DBS	1
Total		94

GPi: globus pallidus internus; STN: subthalamic nucleus; DBS: deep brain stimulation; SCM: sternocleidomastoid; DAM: denervation and myotomy; VIM: ventralis intermedius.

**Table 3 brainsci-12-00881-t003:** Statistical analysis of improvement in TWSTRS and Tsui scores in the DAM group and DBS group.

Group	*p*-Value	Pre-O and Post-O	
		TWSTRS Score	Tsui Score
DAM (*n* = 41)	t (z)	8.762	−5.103
	*p*	0.000 *	0.000 *
Improvement (%)		42.20	45.64
DBS (*n* = 53)	t (z)	−6.308	−6.210
	*p*	0.000 *	0.000 *
Improvement (%)		61.08	63.87
DAM and DBS (*n* = 94)	t (z)	−3.161	−1.796
	*p*	0.002 *	0.072

TWSTRS: the Toronto Western Spasmodic Torticollis Rating Scale; DAM: denervation and myotomy; DBS: deep brain stimulation; Pre-O: preoperative; Post-O: postoperative; *: *p* < 0.05.

**Table 4 brainsci-12-00881-t004:** Relative improvement in TWSTRS and Tsui scores at the last follow-up after DAM surgery.

Factors		*p*-Value	Pre-O TWSTRS Score	Post-O TWSTRS Score	Pre-O Tsui Score	Post-O Tsui Score
Sex	M (25), F (16)	t (z)	−1.572	0.323	−1.029 (z)	−0.111
		*p*	0.124	0.749	0.304	0.912
Age at surgery (years)	≤40 and >40	t (z)	−0.542	−0.898 (z)	−0.242	−0.162
		*p*	0.591	0.369	0.810	0.872
	≤50 and >50	t (z)	0.304	−0.817	−0.675 (z)	−0.613
		*p*	0.763	0.419	0.500	0.542
Disease duration (months)	≤24 and >24	t (z)	−3.800	1.479	−0.745 (z)	−0.728
		*p*	0.000 *	0.145	0.456	0.467
	≤60 and >60	t (z)	−0.352	−0.996	−1.738	−1.219
		*p*	0.727	0.325	0.090	0.230
Follow-up period (months)	≤100 and >100	t (z)	−0.412	1.226	1.096	−1.278 (z)
		*p*	0.683	0.228	0.280	0.201

Pre-O: preoperative; Post-O: postoperative; TWSTRS: the Toronto Western Spasmodic Torticollis Rating Scale; *: *p* < 0.05.

**Table 5 brainsci-12-00881-t005:** Relative improvement in TWSTRS and Tsui scores at the last follow-up after DBS surgery.

Factors		*p*-Value	Pre-O TWSTRS Score	Post-O TWSTRS Score	Pre-O Tsui Score	Post-O Tsui Score
Sex	M (27), F (26)	t (z)	−2.119	−0.810	0.564	−0.405
		*p*	0.034 *	0.418	0.575	0.687
Age at surgery (years)	≤40 and >40	t (z)	−1.225	−1.856	−1.384	−0.272
		*p*	0.221	0.063	0.172	0.786
	≤50 and >50	t (z)	−0.357	−1.353	−2.930	−0.874
		*p*	0.721	0.176	0.003 *	0.382
Disease duration (months)	≤12 and >12	t (z)	0.733	−1.581	0.986	−1.298
		*p*	0.467	0.114	0.329	0.194
	≤24 and >24	t (z)	−0.614	−0.240	1.338	−0.020
		*p*	0.539	0.810	0.187	0.984
	≤60 and >60	t (z)	−1.478	−1.106	−1.726	−0.116
		*p*	0.139	0.269	0.084	0.908
Follow-up period (months)	≤24 and >24	t (z)	−1.620	−0.079	−1.530	−1.428
		*p*	0.105	0.937	0.126	0.153
	≤36 and >36	t (z)	−0.241	−1.114	−1.784	−1.740
		*p*	0.810	0.265	0.074	0.082
	≤48 and >48	t (z)	−1.034	−2.068	−2.892	−3.287
		*p*	0.301	0.039 *	0.004	0.001 *

Pre-O: preoperative; Post-O: postoperative; TWSTRS: the Toronto Western Spasmodic Torticollis Rating Scale; *: *p* < 0.05.

## Data Availability

Not Applicable.

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
