# Peer review of "The Long-Term Efficacy, Prognostic Factors, Safety, and Hospitalization Costs Following Denervation and Myotomy of the Affected Muscles and Deep Brain Stimulation in 94 Patients with Spasmodic Torticollis"

_brainsci, 2022, doi:10.3390/brainsci12070881_

Round 1
Reviewer 1 Report
Straightforward, well written paper. I commend the authors for evaluating relative costs of the surgical approaches. This is important in all countries, regardless of how health costs are managed.
Two moderate concerns:
1. The TWSTRS results should be evaluated separately for the motor severity, pain and disability subsections. The authors acknowledge these different sections in the Discussion, but as far as I could tell, all of the TWSTRS results are reporting the overall total (?). Then, oddly, in the Conclusions, they say that DBS had a stronger effect on QoL and pain than DAM, based on the TWSTRS. So either they are inferring that from the overall total TWSTRS score (which would be obscuring effects specific to the motor severity), or they are drawing that conclusion based on different results with the different parts of the TWSTRS that are not reported earlier in the manuscript.
2. There seems to be an over-emphasis not only on tables (some of which might be more efficiently communicated to the reader with a simple bar plot, e.g. Table 3 at least), but also on reporting results in terms of the test statistics in the tables, rather than the raw data themselves with the test statistics as an adjunct.
Minor comments:
a. Many places in the manuscript they report numbers with 2 digits after the decimal point where that level of accuracy seems inappropriate.
b. reporting p = 0.0000 is inappropriate.
c. their gender mix (52 M, 42 F) is worth commenting on in the Discussion (how does this compare to the overall prevalence of isolated CD, which typically has much higher proportion of female?).
d. because most of the DAM procedures were done before 2012 and most DBS since then, when comparing costs the authors should adjust accordingly (i.e. using equivalent costs at one point in time)
Reviewer 2 Report
- The Author has to mention. How has been the GPi operative localized? which coordinates have been used?
- which setting parameters have been used in stimulation?
- How was the improvement in STN and VIM DBS group comparing with GPi on Dystonia?
Reviewer 3 Report
The organization of this paper is fine and very logically written. However, it does not show the outcomes of different DAM techniques and different DBS targets. I would like to see a comparison of outcomes by treatment technique, as well as the criteria for treatment selection in DAM and DBS.
Reviewer 4 Report
The manuscript of Cui et al “The long-term efficacy, prognostic factors, safety, and
hospitalization costs following denervation and myotomy of the affected
muscles and deep brain stimulation in 94 patients with spasmodic torticollis” is well written and concerns an interesting topic. I have a few specific questions:
1. In the Abstract the authors state the TWSTRS scores were significantly higher in the in the DBS group than the DAM group but this contradicts statement in Results section paragraph 2 “In terms of TWSTRS scores, we found no significant between the two groups before surgery (P=0.650). Perhaps the authors meant to say that the baseline Tsui scores differed between DBS and DAM groups at baseline, as this is reported in Results paragraph 3.
2. Methods page 2: we are told 68 patients received Botox injections. What proportion of patients continued to require Botox injections after DAM or DBS?
3. Page 3 last paragraph. Are the authors certain that 2 patients had cervical dystonia combined with Parkinson’s disease? Limb tremor and stiffness (without bradykinesia) can be seen in dystonic syndromes. Were either of these patients the same as those deemed Tardive as residual effects of neuroleptic medications could simulate Parkinson’s. Diagnostic clarification required here.
4. Results section: please provide the mean percentage improvement values for DAM and DBS for TWSTRS and Tsui in the text and the tables. These are mentioned in the discussion but belong in the Results section.
5. The authors found greater benefit for DBS in those with longer follow up (results page 5 final paragraph). This is consistent with well-described progressive improvement after DBS that has been interpreted as suggestive of long-term adaptive plasticity. This would be worth mentioning in the discussion.
6. Overall DAM was less effective than DBS in terms of TWSTRS outcome. Would this result have been the same with shorter follow-up eg 12 months? It is well described that beneficial effects of DAM may wane over time due to re-innervation, such that long term outcomes may be inferior to those in the shorter term. It would be interesting of the authors had some earlier follow up data to test this hypothesis.
7. The authors mention that higher cost of DBS is a factor in patient selection for DAM vs DBS. To what extent, in this series, did financial status of patients determine the choice of procedure, or was it driven more by the clinical presentation?
8. The authors describe that DBS was more effective than DAM in improving pain and quality of life but make no attempt to explain why. This point is relevant because the improvement in dystonic posture between DAM and DBS, based on Tsui scale, was no different. Some discussion about mechanisms of action and the basis for overall superiority of DBS should be included.
Round 2
Reviewer 1 Report
The authors did not fully address my suggestions in a satisfactor manner, in 2 regards: 1. regarding my initial comment about the gender balance in their study …. they explain the reasons for it in their cover letter but did not make any change i suggested in the Discussion 2. They said they changed a table to a bar plot. But I do not see the bar plot. And oddly, the table that they did change changed in a way OPPOSITE that I suggested (i.e. they replaced actual outcome measures with just the statistics). These still need to be addressed, and I will be willing to review it one more time.
Author Response
Thank you for your comments. The reply is attached.

Reviewer 3 Report
This paper is very logically revised and written. I understand that it is difficult to compare the outcomes of different DAM techniques and different DBS targets due to the small number of cases.
Author Response
Thanks for your critical comments.